# The Prognostic Role and Significance of Dll4 and Toll-like Receptors in Cancer Development

**DOI:** 10.3390/cancers14071649

**Published:** 2022-03-24

**Authors:** Zacharias Fasoulakis, Antonios Koutras, Thomas Ntounis, Vasilios Pergialiotis, Athanasios Chionis, Alexandros Katrachouras, Vasileios-Chrysovalantis Palios, Panagiotis Symeonidis, Asimina Valsamaki, Athanasios Syllaios, Michail Diakosavvas, Kyveli Angelou, Athina A. Samara, Athanasios Pagkalos, Marianna Theodora, Dimitrios Schizas, Emmanuel N. Kontomanolis

**Affiliations:** 11st Department of Obstetrics and Gynecology, General Hospital of Athens ‘ALEXANDRA’, National and Kapodistrian University of Athens, Lourou and Vasilissis Sofias Ave, 11528 Athens, Greece; hzaxos@gmail.com (Z.F.); antoniskoy@yahoo.gr (A.K.); thomasntounis@gmail.com (T.N.); pergialiotis@yahoo.com (V.P.); mdiakosavvas@gmail.com (M.D.); kiv_ang@hotmail.com (K.A.); martheodr@gmail.com (M.T.); 2Department of Obstetrics and Gynecology, Laiko General Hospital of Athens, Agiou Thoma 17, 11527 Athens, Greece; ath.chionis@yahoo.gr; 3Department of Obstetrics and Gynecology, University of Ioannina, University General Hospital of Ioannina, Stavros Niarchos Str., 45500 Ioannina, Greece; alexkatra1994@gmail.com; 4Department of Obstetrics and Gynecology, University of Larisa, University General Hospital of Larisa, Mezourlo, 41110 Larisa, Greece; pampalaios@hotmail.gr; 5Department of Obstetrics and Gynecology, Democritus University of Thrace, Vasilissis Sofias Str. 12, 67100 Alexandroupolis, Greece; simpanthess@gmail.com (P.S.); mek-2@otenet.gr (E.N.K.); 6Department of Internal Medicine, General Hospital of Larisa, Tsakal of 1, 41221 Larisa, Greece; semi_val@hotmail.com; 71st Department of Surgery, Laikon General Hospital, National and Kapodistrian University of Athens, Agiou Thoma Str. 17, 11527 Athens, Greece; 8Department of Surgery, University Hospital of Larissa, Mezourlo, 41110 Larissa, Greece; at.samara93@gmail.com; 9Department of Obstetrics and Gynecology, General Hospital of Xanthi, Neapoli, 67100 Xanthi, Greece; sakispagkalos@gmail.com; 101st Department of Surgery, National and Kapodistrian University of Athens, Laiko General Hospital, 11527 Athens, Greece; schizasad@gmail.com

**Keywords:** endometrial cancer, notch signaling pathway, Dll 4, prognostic role, toll-like receptors

## Abstract

**Simple Summary:**

The aim of this review is to summarize the latest details considering the role of Dll4 in cancer, since recent data report that Dll4 has a major key role in tumor angiogenesis. Moreover, the authors try to seek any correlation between Dll4 and cancer stem cells in tumor development. Considering that cancer stem cells have proven to be implicated in the progression of many cancer types, any impact from Dll4 could lead to the alteration of cancer development. Additionally, the authors make a report on current advantages on immunotherapy and tumor-draining lymph nodes in cancer. Finally, this study analyzes toll like receptors, pattern recognition receptors that are capable of recognizing different molecules and activating different genes. These immunogenetic molecules have remarkable roles including angiogenesis promotion, while their activation can lead to either cancer progression or inhibition, representing a very promising therapeutic alliance for cancer treatment.

**Abstract:**

The Notch signaling pathway regulates the development of embryonic and tissue homeostasis of various types of cells. It also controls cell proliferation, variation, fate and cell death because it emits short-range messages to nearby cells. The pathway plays an important role in the pathophysiology of various malignancies, controlling cancer creation. It also limits cancer development by adjusting preserved angiogenesis and cellular programs. One of the Notch signaling ligands (in mammals) is Delta-like ligand 4 (Dll4), which plays a significant role in the overall malignancies’ advancement. Particularly, sequencing Notch gene mutations, including those of Dll4, have been detected in many types of cancers portraying information on the growth of particular gynecological types of tumors. The current research article examines the background theory that implies the ability of Dll4 in the development of endometrial and other cancer types, and the probable therapeutic results of Dll4 inhibition.

## 1. Introduction

Cancer complexity is disclosed by the tumor’s cell ability to metastasize to close and distant organs. The growth of the vascular system is responsible for the progression of tumor tissues, after escaping from the main tumor. The cells can burst the blood and lymphatic vessels mounting at divergent tissues, circulating the intravascular stream [1]. Angiogenesis, lymphangiogenesis, and perforation play a key role in the initiation and development of cancer. Especially, angiogenesis and lymphangiogenesis are responsible for the creation of the vascular network accountable for confiscating waste products and supplying nutrients, immune cells, and oxygen [2]. There exists a principal alarm particularly in neoplastic and vascularization regions that lead to angiogenesis and lymphangiogenesis. In the course of the latest decades, the unknown healing trails of changing Notch appearance have been questioned because of its role in the formation of the vessel resulting in cancer development.

## 2. Angiogenesis in Cancer

Chemical indicators, emerging from the tumor cells during the development phase, trigger the progression of cancer and metastasis that is subject to angiogenesis processes [1]. According to previous research, tumors do not grow beyond a maximum diameter of 1–2 mm^3^ due to the absence of blood supply, hence causing cells to hinder the development of the tumor [3,4]. As a result, angiogenesis has a major role in the advancement of cancer (Figure 1). Neovascularization for cancer initiation and development consists of four major phases. The first phase involves the local bruise of the outer membrane in the nerves that leads to immediate hypoxia and destruction. Second stage consists of endothelial cells stimulation by relocating angiogenic features. Consequently, the tertiary stage is associated with stabilization and a rapid increase in the endothelial cells while the fourth phase consists of angiogenic aspects that promote vascularization. Besides, Denekamp et al. have proven that, on average, vascular endothelial cell metalates after every 1000 days [5].

### 2.1. Angiogenesis via Notch Stimulation

The stimulation of angiogenesis takes place if the tumor tissues fulfill the requirement of both oxygen and nutrients. Inhibitor and activator chemicals also contribute to the regulation of angiogenesis. Although the down-regulation of non-inhibitors vessel and regulators is essential, upregulation of factors promoting angiogenesis does not satisfy neoplasm vascularization [6].

Presently, the concept of specific pathways with a critical responsibility in vascular function as well as tumor angiogenesis has been widely acknowledged. Cell-to-cell signaling has a specific implication during cancer development via angiogenesis, primarily through the Delta ligand 4 (DII4) of the Notch signaling pathway [7,8,9,10]. Notch pathway activation initiates the consecutive receiver proteolytic cleavages while the proteins of the ligand remain in extracellular fields. Therefore, the intracellular domain and release cleavages get prompted to enter the nuclear cell and alter the gene expression [11].

The Notch gesturing lane enhances cell behavior. It is also fundamental limit in every primary cell-to-cell engagement stage. Besides, it does not take part in the coordination of pathway signaling, which requires the regulation of gene mechanism governing several processes in cell variation. In addition, cell fate has one of the vital roles in diverse procedures of embryogenesis, regeneration of cells and tissues, and organogenesis [12,13]. NECD-NTMIC is associated with singular Notch cellular receptor transmembrane proteins, including an extracellular, an intracellular area, and a transmembrane receptor. Mammals possess four types of NOTCH receptors (1, 2, 3 and 4). There exists endoplasmic reticulum in the Golgi bodies of cells receiving signals, for the receptor processing.

Glycosylation and cleavage play an essential role in the creation of stabilized NCED calcium hetero-dimer that is not often attached to TM-NCID implanted in the membrane. One of the models associated with the renovating enzyme, states how the NECD is often processed and cleaved off from TM-NICD. Consequently, the processed NECD becomes endosome, that is, transferred through the send signal cell membrane. This leads to the reprocessing of the NECD part in the cell plasma and freeing of NICD by γ secretase from TM getting signal cell. NICD section is inserted into the nucleus, as the activation of the CSL transcription factor complex, permits the translocation nucleus, leading to the actuation of target genes (Figure 2, Table 1). The Delta-like ligand and Jagged proteins are the most common examples of Notch agonists [14,15]. Examples of Delta-like protein include 3 and 1 pathway receptors and Drosophila protein as well as Delta mammalian homologs, that functions as a section of ligands for notch 4. Dll4 gene encodes Dll4 of human beings. Besides, Dll4 and Jag1 disclose an extremely discriminating pattern of expression within the vascular endothelium and aggressively developing veins and existing arteries. However, many Notch ligands and receptors demonstrate various kinds of cells. Therefore, they exhibit a fundamental role in angiogenesis promotion [16].

### 2.2. Notch Ligands’ Role in Angiogenesis

Tumor vascular form does not have distinctive characteristics and shows untypical functional and morphological features. Consequently, the tumor diligently enlists the blood vessels by inciting sprouting existing blood vessels that lead to the distribution of nutrients that are necessary in the progression of cancer. Research propose that Notch is the main rise of the angiogenesis regulator, which results from a strictly managed equilibrium among the endothelial and stalk tip cells [17]. The cell tip differentiation responds to factors associated with pro-angiogenic to grow vasculature. In particular, Dll4 manages the exposure of endothelial tip cells. Notch –propitiated inhibition of VEGFR2 supports stalk phenotype to avoid hyper sprouting; therefore, it dominates the vasculature structure. The notch control mechanism of development does not specify the cancer settings [18].

Notch ligands use initiation of Notch gesturing to alter the tumor compartments in malignant cancer cells. From numerous investigations, it is revealed that Notch ligands can prompt gesturing of the Notch system in adjacent cancer cells. In specific, aiming mouse Dll4 in xenograft models decreased the activity of Notch in malignant tumor cells [19]. In glioblastoma, it is demonstrated that cancer cell Notch operation is essential in the endothelium cells intimacy [20,21]. This is exhibited in various cancer subtypes and increases the probability of including receptors and Notch glands. For example, Dll4 transported by endothelium cells triggers the cells of Notch 3 to T-ALL and activates the termination of latency period [22]. The Notch initiation in cancer cells by approaching the blood vessels is also recognized to raise the transportation of trans-endothelial, leading to the spread [23]. The signaling Notch is also activated through the expression of Jag1 by endothelial cells in the pericyte predecessor principal cells to assist in inducing pericyte differentiation [24]. Endothelial cells-expressed-ligands also have a responsibility in controlling cancer stem cell traits. Consequently, Notch signaling controls the survival and differentiation of vascular endothelium through inherent mechanisms of heterotypic relations with cancer. Besides, Notch-Conciliated Shaping controls the resistance of different cancer types. The resistance infiltration consists of immune cells participating in cancer cells and is perceived as the main tumor progression controller.

Notch receptors can also function as independent cancer oncoproteins, limiters, and microenvironment-dependent oncoproteins in the context of distinctive cellular. However, the regulation process depends on the tumor, hence raising questions considering the inference of Notch on therapeutic pathways. The constancy amongst Dll4 and Jag1 has a significant consequence on the blood vessel structure because both of them have different responsibilities in controlling angiogenesis development [25]. The mathematical modeling illustrates that the substantial levels of Jag1 may lead to imperfective and chaotic perfused angiogenesis through the weakening of the stalk or tip phenotype [26]. The high expressions of endothelial cells’ Jag1 increase tumor in the blood vessel system, but the loss of Jag1 meaning in endothelial cells reduces vasculature and growth of tumors [27]. In the management of tip ratio, Notch infers the control leakage breakage from cell latency causes cancer, and the tip cells have an association with the process [28].

Wang et al., studied VEGF and Dll4/Notch pathways in tumor angiogenesis. The purpose of this study was to investigate the expression of these two pathway molecules in ovarian cancer and their possible relationships in carcinogenesis. Twenty-eight (28) specimens of human ovarian carcinoma, 18 of benign ovarian and 20 of healthy ovarian tissues were subjected to immunohistochemical analysis for VEGF, VEGFR1, and VEGFR2, Dll4, Notch1, and Notch3 expression. Microvessel density (MVD) was evaluated by counting the number of CD34-stained microvessels in each pathologic specimen. The authors reported that the expression of VEGF, VEGFR1, Dll4, Notch1, or Notch3 in ovarian tumor tissues was higher than that in normal or benign ovary tissues (*p* < 0.05) while in the tumor tissues, Dll4 was positively correlated with VEGFR1 expression and Notch1 was positively associated with VEGFR2 and MVD. Moreover, VEGFR2 expression was positively associated with ascites and distant metastasis (R = 0.401, *p* = 0.034), with the authors concluding that Dll4 represents a potential biomarker and therapeutic target for ovarian angiogenesis. VEGFR2 is significantly related to ovarian metastasis and invasion [29].

Gastric cancer stem/progenitor cells (GCSPCs) have critical effects on tumor formation and metastasis. The Notch-1 pathway is crucially important to GCSPCs and is regulated by DLL4. Liu et al., reported that DLL4 expression is associated with TNM stage and cancer metastasis, with high amounts of DLL4 leading to poor outcome. DLL4 silencing inhibited the self-renewal ability of GCSPCs and increased their multidifferentiation capacity, resulting in reduced GCSPC ratios. DLL4 knockdown also blocked the Notch-1 pathway, weakening invasion ability and resistance to 5-FU chemotherapy. In vivo, DLL4 silencing inhibited the tumor formation ability of GCSPCs, with the authors resulting that DLL4 affects GCSPC stemness, altering their pathological behavior. DLL4 silencing inhibits GCSPC metastatic potential by impeding Notch-1 signaling pathway activation, indicating that DLL4 may be a new potential therapeutic target [30].

## 3. Dll4 in Cancer Development

Dll4 inhibition regulates cancer stem cells frequency and suspends tumor growth. Dll4 overexpression has proven to be implicated in cancer development by promoting tumor growth. Another study involving 383 patients suffering from human gastric cancer (GC) were analyzed with their tissue samples immersed in immunohistochemical discoloration the appearance of Dll4 to determine the distinguished and undistinguishable gastric tumor stem cells. Fascinatingly, positive Dll4 appearance was meaningly related to improved lymph node metastasis and distal metastasis danger as likened with patients presenting adverse Dll4 appearance. The connection between Dll4 appearance level and the cancer stem cell associated protein Nestin (an angiogenesis indicator of multiplying endothelial cells in colorectal tumor vessels) was also analyzed. Positive appearance of Dll4 was proven to be related to Nestin. The authors concluded that Dll4 is associated with gastric cancer progenitor cells, and its expression influences features linked to the Notch-1 pathway involving tumor formation, growth and development [31]. 

Hu et al., examined the clinical significance of Dll4 in ovarian carcinoma utilizing immunohistochemical peroxidase discoloration in eighty-four patients. Eighty-three percent of cancers had severe histology and ninety one percent had extreme level of histology. 88% of participants had progressive phase ailment and fifty nine percent related ascites. Dll4 was administered endothelial and cancer sections of ovarian cancers and its appearance was not linked to grade and extent of cytoreduction. The authors reported that Dll4 overexpression and suboptimal cytoreduction were self-regulating forecasters of poor survival. Moreover, muzzling Dll4 reaction with Dll4 siRNA repressed explosion of ovarian tumor cells by 2.1-fold associated with the regulation. Moreover, the authors examined the impacts of intervention with restrained Dll4 for 48 h on cell relocation. Immobilized Dll4 bigger the relocation of (murine ovarian endothelial cells) MOECs by 2.7-fold compared to interfered cells (*p* < 0.05) although it showed no impact on VEGF-initiated relocation. The major results of the current research show that Dll4 overexpression was significantly connected with reducing health outcome and survival. Moreover, it predicted the Dll4 reaction to anti-VEGF intervention. The suppression of Dll4 in cancer cells caused by reserve of ovarian tumor development and control of angiogenesis, escorted by initiation of hypoxia in the tumor microenvironment, revealing that Dll4 has a key role in ovarian cancer development and that focusing on Dll4 could improve the effectiveness of ovarian tumor intervention n [32]. In addition to the previous study, Yen et al., reported the use of anti-Dll4 treatment for ovarian cancer by regulating cancer stem cell function and tumor angiogenesis. The authors utilized anti-human Dll4 (OMP-21M18) and anti-murine Dll4 to block Notch signaling and found that anti-Dll4 treatment was broadly efficacious in these ovarian cancer models, significantly inhibiting tumor growth [33].

Hoey et al. investigated the results of Dll4 inhibition in cancer stem cells by creating antibodies (anti-hDll4 21M18) selectively aiming at Dll4 in the cancer or in the host vasculature and stroma in xenograft models derived from primary human tumors. Each antibody was proven to inhibit cancer development and that the grouping of two antibodies was even more effective. Administration of anti-human Dll4 reserved the administration of Notch target genes reducing proliferation of cancer cells, reduced cancer stem cell frequency, and deregulated angiogenesis by aiming at Dll4 in the vasculature [8]. The effect of monoclonal antibody (MEDI0639) that selectively binds to Dll4 was also examined in small-cell lung tumor. Tumor stem cells are responsible for the high metastatic profile and rapid frequency of many cancer types. In half of the patients that MEDI0639 was administered, the tumor stem cells frequency was suppressed while 25% of the patients demonstrated >50% reduction of the tumor [34]. In a more recent study, tumor metastasis was also reported to be altered by deregulation of Dll4. Lewis Lung Carcinoma (LLC) cells were used to study tumor metastasis in vivo, by endothelial-specific Dll4 loss-of-function. Cancer stem cells were apparently reduced and hypoxia was increased in the tumor that led to an increase in tumoral blood vessel density, but with neo-vessels poorly perfused, with increased leakage and reduced perivascular maturation. Number and burden of macro-metastasis was significantly reduced and the tumor growth was suspended [35].

Yen et al., studied the activity of targeting DLL4 in tumor cells with an anti-human Dll4 antibody and in the host stroma/vasculature with an anti-mouse Dll4 antibody. The combination of these antibodies was efficacious in a broad spectrum of pancreatic tumor xenografts and showed additive antitumor activity together with gemcitabine. Treatment with either human or mouse anti-Dll4 delayed pancreatic tumor recurrence following termination of gemcitabine treatment, and the two together produced an additive effect, suggesting a novel therapeutic approach for pancreatic cancer treatment through antagonism of DLL4/Notch signaling [36].

Zohny et al. have reviewed the oncogenic function of Notch ligands and receptors in different breast cancer subtypes. Notch 1 has an oncogenic function in estrogen receptors (ER) luminal cell lines, tumor negative breast cancer (TNBC) and in invasive ductal carcinoma. Even though the role of Notch 2 remains ambiguous, similarly to Notch 1, represents an oncogenic factor in HER2 and basal subtype invasive ductal carcinoma (IDC) breast cancer, while it is a tumor-suppressor in ER+ luminal and TNBC cell lines. Moreover, Notch 3 is an oncogenic factor for ER+ and HER2+ human patients, but a tumor suppressor for TNBC cell lines and ERBB2 basal tumor cells. Furthermore, Notch 4 is an oncogenic factor for TNBC human breast cancer. Jag1,2 and Dll1 all act as oncogenic factors in Luminal and TBC cell lines, while Dll4 has a wide oncogenic function in different breast cancer subtypes [37].

## 4. Cancer Stem Cells and Dll4 Expression in Endometrial Cancer

Adult stem cells are identified in various types of mature tissue including normal endometrium and endometrial tumor. Menstrual blood-derived stem cells are called endometrial regenerative cells while gene mutations of these stem cells proven to be able to create cancer stem cells. More specific, Kato et al., presented the function of stem cells in endometrial tumor where stem cells identified in the cancerous tissues revealed specific characteristics including reduced expression of differentiation markers, extended repopulating specifications, self-renewal abilities, enhanced metastatic tendency and increased tumorigenicity revealing their key function in endometrial tumor development [38]. Fasoulakis et al. reported that Dll4 is overexpressed in endometrial cancer cells and vasculature and is also elevated in the plasma of a fraction of patients before surgery [39].

The Notch signaling pathway and especially the Delta gene have been found to exist in uterine endometrium. Mazella et al. revealed that the human endometrial cells articulated Dll4 in a design known as spatiotemporal. Immunohistochemistry educations demonstrated the cytoplasm and membrane discoloration with apical localization in the luminal and glandular epithelium and modest diffuse discoloration in the cytoplasm present in the stromal cells while Western spot examination displayed a common scope of the endometrial Dll4 to that in the human umbilical endothelial cells. The placement of Dll4 mRNA in human endometrial cells was determined to be administered in large variations in the glandular epithelium, raised in the proliferative and early productory endometrium. However, the authors found that the Dll4 and mRNA was less in endometrial had no relation with the menstrual cycle. The author failed to study the effect of hormones. In glandular cells, estradiol had little effect, and medroxyprogesterone acetate decreased mRANs. Relaxin induced the Dll4 mRNA. In stromal cells, both estradiol and medroxyprogesterone acetate decreased the Dll4 mRNA [40].

During the past decade, studies have proven that Dll4 happens to encourage explosion and sustain the stem cells through angiogenic, but also non-angiogenic associated devices. Badenes et al. studied the function of Notch ligands and the impact of a Dll4 knockout in colorectal cancer, which led to positive cancer stem cell density accompanied by improved tumor epithelium variation [40]. Another study proved that Dll4 antibodies were able to suppress tumor stem cells in a Small-cell lung cancer subpopulation promoting the importance of Dll4 antibodies in cancer treatment [36]. Other studies have also reported that Dll4 blockage is correlated to inhibition of tumor growth including ovarian, gastric and lung cancer [8,33,41,42,43].

MEDI0639 is an investigational human therapeutic antibody that targets Dll4 to inhibit the interaction between Dll4 and Notch1. The antibody cross-reacts to cynomolgus monkey but not mouse species orthologues. In vitro MEDI0639 inhibits the binding of Notch1 to Dll4, interacting via a novel epitope that has not been previously described. Binding to this epitope translates into MEDI0639 reversing Notch1-mediated suppression of human umbilical vein endothelial cell growth in vitro. MEDI0639 administration resulted in stimulation of tubule formation in a three-dimensional (3D) endothelial cell outgrowth assay, a phenotype driven by disruption of the Dll4-Notch signaling axis. In contrast, in a two-dimensional endothelial cell–fibroblast coculture model, MEDI0639 is a potent inhibitor of tubule formation. In vivo, MEDI0639 shows activity in a human endothelial cell angiogenesis assay promoting human vessel formation and reducing the number of vessels with smooth muscle actin-positive mural cells coverage. Collectively, the data show that MEDI0639 is a potent modulator of Dll4-Notch signaling pathway [44].

The Notch signaling pathway has been proven to be involved in a crosstalk with WNT signaling. Abnormal activation of WNT signaling has been reported in the majority of type-1 endometrial cancer cases with β-catenin mutations in 20–25% of cases. Fatima et al. discussed the Wnt-activating mechanisms in endometrial cancer and reviewed the current advances in anticancer therapy. Given the current lack of therapeutic solutions for advanced and recurrent endometrial cancer, resent evidence support the role of Wnt signaling at early stages of endometrial tumorigenesis, The authors supported that Wnt signaling represents a promising intervention for targeted therapies in endometrial cancer patients. Various inhibitors targeting different molecules of this pathway have been developed including Medroxyprogesterone Acetate (MPA, Levonorgestrel Intrauterine Device, DKN-01 is a humanized monoclonal antibody (Mab) targeting Dickkopf-1 (DKK1), Porcupine Inhibitor, OMP-54F28 -a fusion protein consisting of the extracellular ligand-binding domain of Fzd8 and a human immunoglobulin G1 (IgG1) Fc domain-, Niclosamide, PRI-724 and ICG-001, Salinomycin, Curcumin and miRNA treatment, though only a few studies have addressed the effects of Wnt inhibitors in endometrial cancer and they are still at an early phase and far away from clinical trials [45].

A simultaneous blockade of VEGF/VEGFR and DLL4/Notch signaling pathways leads to more potent anti-cancer effects by synergistic anti-angiogenic mechanisms in xenograft models. A bispecific antibody targeting VEGF and DLL4 (ABL001/NOV1501/TR009) demonstrates more potent in vitro and in vivo biological activity compared to VEGF or DLL4 targeting monoclonal antibodies alone and is currently being evaluated in a phase 1 clinical study of heavy chemotherapy or targeted therapy pre-treated cancer patients (ClinicalTrials.gov Identifier: NCT03292783). However, the effects of a combination of ABL001 and chemotherapy on tumor vessels and tumors are not known. Hence, the effects of ABL001, with or without paclitaxel and irinotecan were evaluated in human gastric or colon cancer xenograft models. The combination treatment synergistically inhibited tumor progression compared to each monotherapy. More tumor vessel regression and apoptotic tumor cell induction were observed in tumors treated with the combination therapy, which might be due to tumor vessel normalization. Overall, these findings suggest that the combination therapy of ABL001 with paclitaxel or irinotecan would be a better clinical strategy for the treatment of cancer patients [46].

Chiorean et al. studied the Enoticumab (REGN421), a human IgG1 monoclonal antibody that binds human Dll4 and disrupts Notch-mediated signaling, in order to determine the safety, dose-limiting toxicities (DLT), pharmacokinetics (PK), and recommended phase II dose (RP2D) of enoticumab. Enoticumab was administered intravenously in 53 patients with the most frequent adverse events (AE) being fatigue, nausea, vomiting, hypertension, headache, and anorexia. Brain natriuretic peptide increase, troponin I increase, right ventricular dysfunction and pulmonary hypertension, and left ventricular dysfunction and pulmonary hypertension were reported in four patients while Enoticumab was characterized by nonlinear, target-mediated PK, and had a terminal half-life of 8 to 9 days. The authors reported that Enoticumab was tolerated, and that good response was noted for both ovarian cancer and other solid tumors [47].

## 5. Immunotherapy and Tumor-Draining Lymph Nodes

Immunotherapy is mainly specialized in treatments that involve solid tumors which are characterized by the instability of the microsatellite. MSI-high endometric cancer is one such example. Nonetheless, the anti-PD-1 monotherapautic procedure is unproductive because the Uterine Serous Carcinomas (USC) are in the group of p53+ including pMM/MSS [40]. Makker et al. reported data of lenvatinib and pembrolizumab combination in metastatic and recurrent (>1 line) endometric cancer. According to the author, the data revealed that the partial and complete response totaled 63.6% of the patients with MSI-H/dMMR (*n* = 11) compared to 38% in MSS/dMMR, represented by (*n* = 94). The MSS/dMMR population experienced a response in 6 months in a sample of 25 cases (69%). The response rate is indicative of an unprecedented USC response rate of 50%. The FDA approved combining lenvatinib and pembrolizunab in October 2019, for both microsatellite-stable endometric cancer and USC. The results used to deduce the combination and prescribe for a particular form of cancer were gathered from phase II single-armed test [44]. The results that would constitute a confirmatory stage III are expected at the ensuing meeting involving the Gynecologic Oncology (SGO). To date, ongoing trials are testing novel therapeutic protocols for patients with advanced or recurrent endometrial cancer, including USPC. The basis for the investigations is limited to strategies in progressive or recurrent endometric cancer and not assessing its benefit to USC [48,49,50,51].

The agents tested included durvalumab, pembrolizumab, atezolizumab and nivolumab, whether combined to ipilimumab or not. Moreover, several tests are being carried out to determine the various immune therapeutical agents associated with endometric cancer. For instance, the RUBY/ENGOT-en 6 stage III trials investigate the result of the addition of dostarlimab (TSR-042), an antibody that is humanized to a chemotherapy that is platinum based. Other trials that explore the role of immunotherapy in endometrial cancer are LEAP/ENGOTen9 and AtTend/ENGOTen 7. Currently, other different studies are also investigating immunotherapeutic agents (e.g., trastuzumab, SYD985), multi-kinase TKI (e.g., lenvatinib, TKI258), PIK3CA inhibitors (e.g., copanlisib, XL147 (SAR245408)), and PARP inhibitors [52,53]. The experimentation involving HER2 in targeting has revealed positive results [41,42]. This means that HER2 would be one of the most important immunotherapeutic targets in USC. Concerning antibody conjugates, several clinic and preclinic studies have evaluated exhaustively their characteristics. For example, SYD985 targets HER2 which targets a form of antibody drugs conjugate (ADC) comprising of trastuzumab, which is connected to duocarmycin. The component is considered an extremely potent alkylating agent of DNA [54]. In preclinical tests against T-DM1, SYD985 showed high reactivity against the USC primary cell which have strong lines (3+) as well as moderate (1+/2+) line, as an expression of HER2. Based on the outcomes of experiments, SYD985 recorded 10- to about 70-fold times more effective than TDM 1 and active against the USC unlike T-DM1.This demonstrates the heterogeneous HER2 expression. Concerning a trial (in phase I) for the HER2 expressing cancers, patients with endometrial carcinoma recorded a 39% response (phase II ongoing) [44,45,46,47,48,49,50,51,52].

Neratinib and afatinib are irreversible molecular inhibitors of HER2, EGFR and HER 4 approved by the FDA to treat EGFR-positive non-small and squamous lung cancer and HER2 positive breast cancer, which both demonstrate a significant activity against primary HER@-amplified cell lines and xenografts. The study is ongoing in stage II trial based on HER2+ USC. In both locally and advanced breast cancer, Dual anti-HER2 inhibition is considered established therapy, another area of active exploration in Her2- expressing endometrial cancer [52,54,55]. 

The combination of trastuzumab with pertuzumab (an acculturated HER2 monoclonal counter acting agent that forestalls receptor dimerization) has shown antitumor results in USC cell lines [48]. Combined treatment was seen to fundamentally increase drug related cytotoxicity, even in low HER2-expressing cells [48]. Since it has been accounted for that an enormous number of USC shows modifications in PI3K pathway-related genes, a few PI3K/AKT/mTOR inhibitors have been examined against essential USC cell lines and xenografts [49,50]. Preclinical investigations of AZD8055 (mTORC 1/2 inhibitor), GDC-0980 (inhibitor of class one PI3K and mTORC 1/2), and GDC-0032 (taselisib, PIK3CA inhibitor) have shown promising results [43,54,55].

Moreover, recent data have revealed a well-tolerated and extremely synergistic role of PIK3CA taselisib combined to neratib pan-Her. The combination prevented resistance in preclinical USC models, and led to substantial cancer regression in large xenografts, previously resistant to single-agent PIK3CA or pan-Her inhibition. These preclinical results suggest that combination regimens using highly targeted drugs may be of great benefit while synergistic combinations could induce more durable clinical responses in USC [56,57,58,59].

Trop-2, a transmembrane glycoprotein upregulated in all gynecological cancer types was detected in 95.1% of USC samples. Recent studies evaluated the role of sacituzumab govitecan in Trop2 overexpressing USC cell lines and it was found to be highly active against USPC overexpressing Trop-2 and proved to be of great promise against multiple chemotherapy-resistant human tumors [59,60,61].

Ongoing trials are currently investigating the activity of maintenance niraparib in stage III–IV or platinum-sensitive recurrent USC, the activity of maintenance olaparib in Stage III–IV or platinum-sensitive recurrent endometrial carcinoma, and several olaparib-based combinations in unselected uterine cancer subtypes [52].

The addition of the checkpoint inhibition of the immune system to the PARP seeks to utilize the effects of immunomodulatory PARP inhibitors. This section is still under studies in recurrent endometrial cancer. Relating to the immunity checkpoint blockade (ICB), the therapeutic response has been altered, but it is still limited to primary and secondary resistance. The resistance is facilitated by the lack of T-cells sensitive to the tumor. Ways to overcome the situation have proved ineffective considering the strategies emphasizing on the microenvironment of the tumor rather than the TDLN [60,61,62,63,64,65,66,67,68,69].

On the other hand, CTLA-4 blockade TDLN have long since been implicated due to its perceived mechanism-of-action involving T cell priming with recent evidence showing that TDLN are vital for the efficacy of PD-1 blockade. TDLN are under target that tumors are developed to create a mechanism of defense system that ensures priming of T-cells that are antitumor. Francis et al., reported that CTLA-4 and/or PD-1 administration results to easy access to TDLN. Moreover, the author adds that tumor protection can also be achieved by lymph draining to a similar lymph node and creating an equal protection mechanism to tumor in its microenvironment. This is achieved through the administration of ipsilateral on a place different from where the tumor is located. Nonetheless, systemic treatment using ICI in the early stages of cancer results to high toxicity levels. Thus, lower administration of doses is recommended but efficacy should be maintained to ensure TME and TDLN are under target [70,71,72,73,74,75,76].

Based on patients suffering from advanced melanoma, administration of CpG can lead to increased T cell infiltration overcoming previous resistance to PD-1 blockade, providing systemic tumor control. Oncolytic virus therapies, such as local treatment with the oncolytic Herpes Simplex virus Talimogene laherperepvec (T-VEC), are similar to local injection with TLR agonists. They are called engineered virus because they constantly replicate the cells that are in tumor and secrete cytokine. The T-VEC cell, which is part of the Oncolytic viruses are capable of inducing the responses of the immune system through a process called death induction. The process may result into DCs being activated and may drain into lymph nodes to a TDLN environment where resident of lymph node is activated, thereby giving room to the occurrence of T-cell priming. Thus, more studies on TDLN are of great necessity in order to establish the correct dosage, treatment combinations and surgery options [75,76,77,78,79,80,81].

## 6. Toll-like Receptors in Endometrial Cancer

There are two first-line defensive mechanisms in the immune system. Innate and adaptive immunity. In the first one the components of microorganisms bind to Toll-like receptors (TLRs) and activate the inflammatory response, promoting elimination of the invading microorganisms. The second one is activated when the first line is overpassed with dendritic cells activating T and B lymphocytes, producing antibodies and NK (natural killer) cells that destroy infected-by-invaded pathogenic microorganism cells. TLRs are members of the interleukin-1 receptor family [82] that trigger a signal transduction pathway to initiate gene translation. By recognizing the molecular structure of foreign pathogens, the produced protein innates immune response and develops antigen-specific acquired immunity [83,84].

There are 10 functional TLRs that have been identified in humans (TLR1-TLR10). TLR1, TLR2, TLR4, TLR5, TLR6 and TLR10 are expressed on the cell’s surface and when activated, they transfer to phagosomes. TLR3, TLR7, TLR8 and TLR9 are expressed in the endosomes or the endoplasmic reticulum with ligand-binding domains. TLRs demonstrate different ligand binding and expression patterns, while they target different genes. When a TLR binds to its ligand, multiple defensive genes are expressed. These include cytokines, chemokines, antimicrobial peptides, costimulatory molecules, major histocompatibility (MHC) molecules and other effectors, so as to adequately attack the invading pathogen [85,86].

TLRs are pattern recognition receptors (PRRs), capable of detecting numerous different molecular structures. In bacteria and viruses, TLR3 recognizes double stranded DNA, TLR4 detect lipopolysaccharides, TLR5 flagellin, TLR7 and TLR8 recognizes single stranded viral RNA and TLR9 unmethylated CpG (Cysteine–Phosphate–Guanine) sites of DNA [79]. Moreover, TLRs are also characterized as transmembrane proteins since they can detect pathogen associated molecular patterns (PAMPs). TLRs have a variety of functions in tissue homeostasis, regulation of cell death and survival. The have also been implicated in both autoimmune diseases and cancer development—tumorigenesis, such as kidney clear cell carcinoma (KIRC). Zou et al., reported that the occurrence and development of KIRC are closely related to TLRs, and TLRs have the potential to be early diagnostic and prognostic biomarkers of KIRC [87,88]. TLR can regulate cell proliferation and survival, and benefit tumor cells through environmental changes beneficial for inflammatory response, angiogenesis and cell death [89].

Considering angiogenesis, TLRs appear to play a major key role in tumor development. Vascular endothelial growth factor (VEGF) is secreted by tumor cells, immune cells and cancer associated fibroblasts (CAFs) and create high interstitial pressure and hypoxia, which stimulates additional VEGF production due to its permeability. Except for PAMPs, TLRs also recognize damage-associated molecular pattern (DAMP), nucleic acid proteins that are released during cell death. DAMP activation of TLRs initiates signaling cascades that release cytokines and chemokines from cancer cells [81,82]. This activation leads to secretion of additional cytokines, proangiogenic mediators and growth factors that can facilitate tumor growth. Fazeli et al. reported the in vitro expression of TLRs in the female productive tract from immunohistochemical testing hysterectomy specimens due to benign diseases. TLR1, TLR2, TLR3, TLR5 and TLR6 were recognized in the epithelia of many female reproductive tract tissues, with TLR4 being present only in the endocervix, endometrium and fallopian tubes [90,91].

TLRs are expressed in human endometrium and could have an important role in pathogenesis of endometrial cancer. Allhorn et al. reported different TLR3 and TLR4 protein levels (mostly located in glandular and luminal epithelium) in the different endometrial tissues including samples from normal menstrual cycles, endometriosis, postmenopausal endometrium, endometrial hyperplasia and endometrial carcinoma [92]. There were no significant differences between TLR3 and TLR4 mRNA levels during the menstrual cycle, but their levels decreased significantly in proliferative endometrium. Ectopic endometriotic lesions revealed considerable increase of TLR3 and TLR4 mRNA compared to normal tissue. Endometrial hyperplasia and adenocarcinoma were reported to have lower levels of receptors compared to postmenopausal endometrium. The lowest expression levels were in poorly differentiated (grade 3) endometrial adenocarcinoma. Ashton et al. found that TLR9 polymorphisms were protective against endometrial cancer [93], while Rajput et al. reported paclitaxel-dependent activation of TLR4 is more relevant to breast cancer chemoresistance. The authors reported that paclitaxel not only destroys tumor cells but also enhances their survival through TLR-4 pathway activation [94,95].

## 7. Conclusions

Dll4 reveals a major key role in endometrial cancer formation while it seems to have a critical role in both tumor angiogenesis and cancer stem cells activation. Immunotherapies represent a promising novel therapy however, there are still ongoing research that will lead to important information considering the appropriate protocols for the different types of cancer. Taking into consideration the presence and the role of tumor stem cells in endometrial formation, the implication of Dll4 gene in endometrial cancer development and the interaction between them, where Dll4 blockage has proven to be correlated to cancer stem cell inhibition and suspension of tumor development, an interaction between Dll4 and cancer stem cells in endometrial cancer seems quite possible, however, more research is predominant to reach safe conclusions.

## Figures and Tables

**Figure 1 cancers-14-01649-f001:**
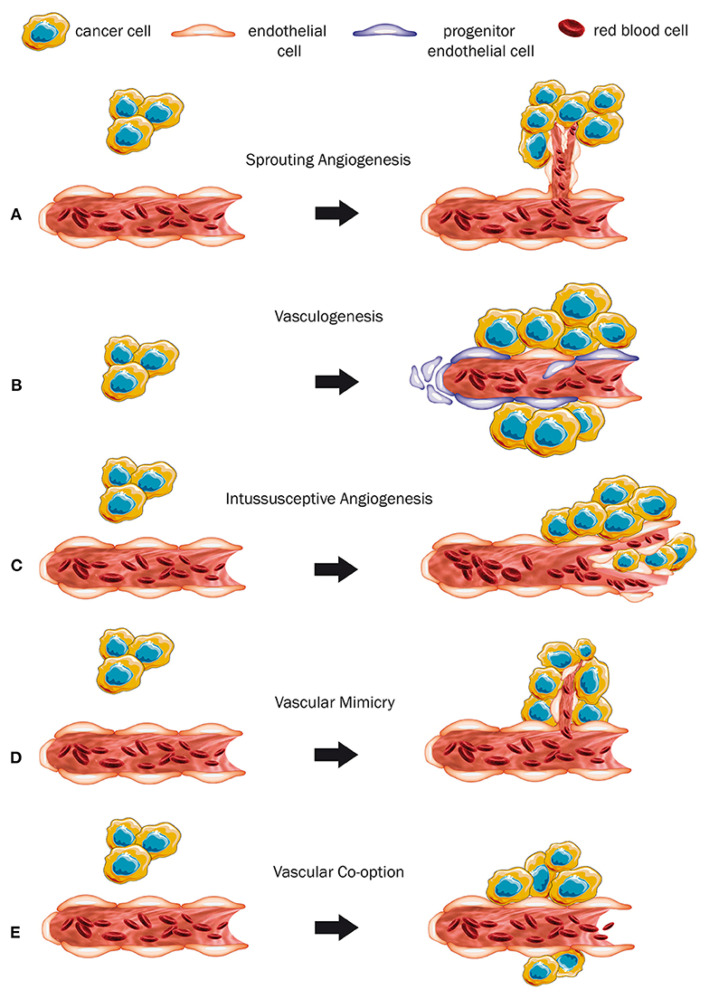
Different neovascularization types due to cancer. (**A**) Sprouting angiogenesis: growth of new capillary vessels out of preexisting ones. (**B**) Formation of primitive vascular structures during embryogenesis via the differentiation of endothelial precursor cells. (**C**) A dynamic intravascular process capable of dramatically modifying the structure of the microcirculation. (**D**) Formation of vascular structures by cancer cells, allowing to generate a channel-network able to transport blood and tumor cells. (**E**) Mechanism in which tumors obtain a blood supply by hijacking the existing vasculature and tumor cells migrate along the vessels of the host organ. (Permission by Haas G, Fan S, Ghadimi M, De Oliveira T, Conradi LC. Different Forms of Tumor Vascularization and Their Clinical Implications Focusing on Vessel Co-option in Colorectal Cancer Liver Metastases. Front Cell Dev Biol. 12 April 2021;9:612774. doi: 10.3389/fcell.2021.612774. PMID: 33912554; PMCID: PMC8072376).

**Figure 2 cancers-14-01649-f002:**
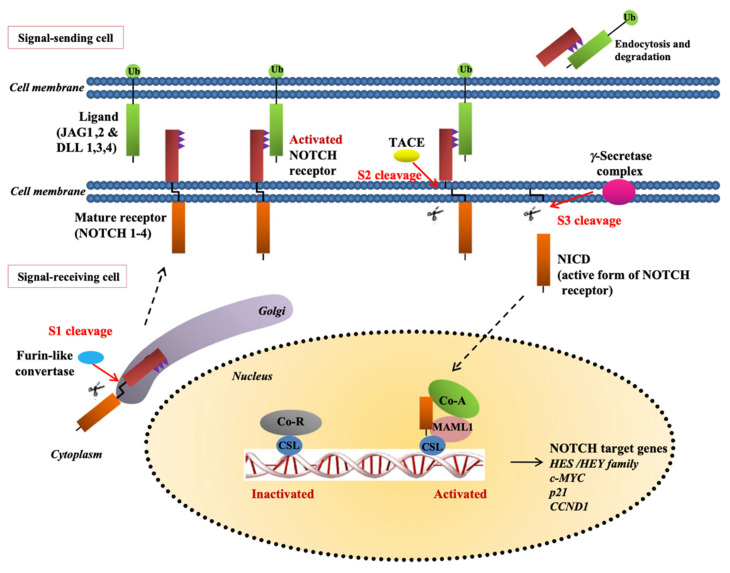
The notch pathway. Construction of the receptor in the endoplasmic reticulum and Golgi is followed by cleavage of NECD from TM-NICD with the converting enzyme TACE. The processed NECD is endosome-transported in the-signal sending-cell plasma membrane where it is recycled. γ-secretase releases NICD from TM in the signal-receiving cell and the NICD part enters nucleus and with the activation of CSL transcription factor complex allows nuclear translocation resulting in activation of the canonical notch target genes (Permission by Yap LF, Lee D, Khairuddin A, Pairan MF, Puspita B, Siar CH, Paterson IC. The opposing roles of NOTCH signaling in head and neck cancer: a mini review. Oral Dis. October 2015;21(7):850-7. doi: 10.1111/odi.12309. Epub 2015 Jan 29. PMID: 25580884).

**Table 1 cancers-14-01649-t001:** Notch signaling pathways target genes.

Role	Target Gene
Apoptosis	NFKB1, CDKN1A, CFLAR, IL2RA
Cell cycle regulators	CCND1, P21, P27, IL2RA
Cell proliferation	P21, P27, ERBB2, FOSL1, IL2RA
Cell differentiation	DTX1, HES6, PPARG
Neurogenesis	HES1, HEY1, HEY2
Transcription	NFKB1, NR4A2, PPARG, STAT6, DTX1, HES1, HES6, HEY1, HEY2, FOS, FOSL1
Unspecified	CD44, CHUK, PTCRA, LOR, MAP2K7, PDPK1, MGC61598, HES5, IFNG, IL 17B, IVL, KRT1, KRT10, KRT14, KRT5, LOR

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
