# Peer review of "The Prognostic Role and Significance of Dll4 and Toll-like Receptors in Cancer Development"

_cancers, 2022, doi:10.3390/cancers14071649_

Round 1

Reviewer 1 Report

All my concerns were addressed in the revised manuscript.

Author Response

Dear reviewer 1,

Thank you for your time and for reviewering our manuscript titled "The Prognostic Role and Significance Of DLL4 In Endometrial Cancer Development". There have been some changes considering the second reviewer's comments.

Reviewer 2 Report

In their manuscript “ The Prognostic Role and Significance Of DLL4 In Endometrial Cancer Development" Zacharias et al. intended to explore the underlying hypothesis that implicates D114 in the development of endometrial cancer, as well as the potential therapeutic effects of Dll4 inhibition.

This is a brief overview that covers a wide range of topics, including

  1. the mechanism of angiogenesis development in cancer.
  2. role of Notch signaling in the angiogenesis process.
  3. Why targeting D114 could be advantageous for slowing tumor development.

In my previous evaluation, I listed a few improvements that I did not see incorporated into the revised version.

There are a number of substantial issues that need to be resolved, listed below.

  1. Since this overview is not confined to the development of endometrial cancer, the title can be generalized to include a discussion of cancer in total.
  2. If that is the case, I would suggest the authors to include a couple of more pieces of research that the authors may have overlooked in their review.

Wang, H., Huang, X., Zhang, J., Shao, N., ong Chen, L., Ma, D., & Ji, C. (2014). The expression of VEGF and Dll4/Notch pathway molecules in ovarian cancer. Clinica chimica acta436, 243-248.

Liu, Z., Fan, F., Wang, A., Zheng, S., & Lu, Y. (2014). Dll4-Notch signaling in regulation of tumor angiogenesis. Journal of cancer research and clinical oncology140(4), 525-536.

Yen, W. C., Fischer, M. M., Hynes, M., Wu, J., Kim, E., Beviglia, L., ... & Hoey, T. (2012). Anti-DLL4 has broad spectrum activity in pancreatic cancer dependent on targeting DLL4-Notch signaling in both tumor and vasculature cells. Clinical Cancer Research18(19), 5374-5386.

Zohny, S. F., Zamzami, M. A., Al-Malki, A. L., & Trabulsi, N. H. (2020). Highly expressed DLL4 and JAG1: their role in incidence of breast cancer metastasis. Archives of medical research51(2), 145-152.

Etc.

  1. Since the authors are not specifically covering endometrial cancer in this review article, they should include current research that has primarily focused on D114 inhibition and has shown the effect on different cancer stages. Such as,

Jenkins, D. W., Ross, S., Veldman-Jones, M., Foltz, I. N., Clavette, B. C., Manchulenko, K., ... & Barry, S. T. (2012). MEDI0639: a novel therapeutic antibody targeting Dll4 modulates endothelial cell function and angiogenesis in vivo. Molecular cancer therapeutics11(8), 1650-1660.

Yeom, D. H., Lee, Y. S., Ryu, I., Lee, S., Sung, B., Lee, H. B., ... & You, W. K. (2021). ABL001, a Bispecific antibody targeting VEGF and DLL4, with chemotherapy, synergistically inhibits tumor progression in xenograft models. International Journal of Molecular Sciences22(1), 241.

Chiorean, E. G., LoRusso, P., Strother, R. M., Diamond, J. R., Younger, A., Messersmith, W. A., ... & Jimeno, A. (2015). A phase I first-in-human study of enoticumab (REGN421), a fully human delta-like ligand 4 (Dll4) monoclonal antibody in patients with advanced solid tumors. Clinical Cancer Research21(12), 2695-2703.

New Suggestion:

  1. Recently, a very fine review was written, which might be used to increase the quality of this article.

“Fatima, I.; Barman, S.; Rai, R.; Thiel, K.W.W.; Chandra, V. Targeting Wnt Signaling in Endometrial Cancer. Cancers 202113, 2351. https://doi.org/10.3390/cancers13102351”

  1. I would suggest the authors to pay special attention to the abbreviation of s Delta-like ligand 4 as it is different in different places. (like DLL4, D114, or DII4), Please rectify which one is correct.

I would recommend the article for publication after the above major corrections.

Author Response

Dear reviewer,

Thank you for your time on reviewing our manuscript. We have made the changes listed bellow:

  1. The title is now generalized to include a discussion of cancer in total.
  2. We have included the following articles according to your suggestions:

Wang, H., Huang, X., Zhang, J., Shao, N., ong Chen, L., Ma, D., & Ji, C. (2014). The expression of VEGF and Dll4/Notch pathway molecules in ovarian cancer. Clinica chimica acta, 436, 243-248.

Liu, Z., Fan, F., Wang, A., Zheng, S., & Lu, Y. (2014). Dll4-Notch signaling in regulation of tumor angiogenesis. Journal of cancer research and clinical oncology, 140(4), 525-536.

Yen, W. C., Fischer, M. M., Hynes, M., Wu, J., Kim, E., Beviglia, L., ... & Hoey, T. (2012). Anti-DLL4 has broad spectrum activity in pancreatic cancer dependent on targeting DLL4-Notch signaling in both tumor and vasculature cells. Clinical Cancer Research, 18(19), 5374-5386.

Zohny, S. F., Zamzami, M. A., Al-Malki, A. L., & Trabulsi, N. H. (2020). Highly expressed DLL4 and JAG1: their role in incidence of breast cancer metastasis. Archives of medical research, 51(2), 145-152.

Jenkins, D. W., Ross, S., Veldman-Jones, M., Foltz, I. N., Clavette, B. C., Manchulenko, K., ... & Barry, S. T. (2012). MEDI0639: a novel therapeutic antibody targeting Dll4 modulates endothelial cell function and angiogenesis in vivo. Molecular cancer therapeutics, 11(8), 1650-1660.

Yeom, D. H., Lee, Y. S., Ryu, I., Lee, S., Sung, B., Lee, H. B., ... & You, W. K. (2021). ABL001, a Bispecific antibody targeting VEGF and DLL4, with chemotherapy, synergistically inhibits tumor progression in xenograft models. International Journal of Molecular Sciences, 22(1), 241.

Chiorean, E. G., LoRusso, P., Strother, R. M., Diamond, J. R., Younger, A., Messersmith, W. A., ... & Jimeno, A. (2015). A phase I first-in-human study of enoticumab (REGN421), a fully human delta-like ligand 4 (Dll4) monoclonal antibody in patients with advanced solid tumors. Clinical Cancer Research, 21(12), 2695-2703.

“Fatima, I.; Barman, S.; Rai, R.; Thiel, K.W.W.; Chandra, V. Targeting Wnt Signaling in Endometrial Cancer. Cancers 2021, 13, 2351. https://doi.org/10.3390/cancers13102351”

3.The Dll4 abbreviation is now corrected on the whole manuscript.

Round 2

Reviewer 2 Report

The manuscript is now well written and can be considered for publication.

Author Response

Thank you for your time on reviewing our manuscript. 

This manuscript is a resubmission of an earlier submission. The following is a list of the peer review reports and author responses from that submission.

Round 1

Reviewer 1 Report

Thank you for allowing me to review the manuscript entitled "The Prognostic Role and Significance Of DLL4 In Endometrial Cancer Development" by Zacharias et al.

This is a review article on the role of DLL4 In Endometrial Cancer Development.

The majority of the review is foundation level and does not provide a comprehensive review. Data on DLL4 in endometrial cancer are scarce and therefore, which also eliminates the necessity of a review article.

Reviewer 2 Report

The Notch signalling pathway plays an important role in cancer initiation and progression. The Notch receptor and one of its ligands (Delta-like ligand [DLL]4) are found to be mutated in many types of cancers, particularly in gynaecological cancers. In this manuscript, Zacharias et al. attempted to review the current literature on the prognostic role and significance of DLL4 in endometrial development. Although the topic is interesting and topical, the review article is unclear, less informative and lacks focus. Further, it suffers from typos and grammatical errors. There is no or very little information on the prognostic role of DLL4 and its significance in endometrial cancer development is discussed only in section 3.2. This review can be improved by extensive re-writing.

Reviewer 3 Report

Thank you for allowing me to review this paper. The Authors evaluated the role of  DLL4 in EC. The paper is well-written and interesting. 

  • I suggested the authors add more data regarding other potential targets in non-endometrioid EC. In this paper (PMID: 33934848) the Rare Tumor Working Group evaluated the role of immunotherapy (in MMMRd EC), immunotherapy plus TKI (in p53abn), and other novel agents in serous endometrial cancer. I think that adding those details would improve the discussion
  • It would be interesting to speculate about the role of those new therapies in patients detected with low volume nodal disease (PMID: 30833134) and their potential application in node-positive patients (PMID: 32646864)

Reviewer 4 Report

In their manuscript “ The Prognostic Role and Significance Of DLL4 In Endometrial Cancer Development" Zacharias et al. intended to explore the underlying hypothesis that implicates D114 in the development of endometrial cancer, as well as the potential therapeutic effects of Dll4 inhibition.

This is a brief overview that covers a wide range of topics, including

  1. the mechanism of angiogenesis development in cancer.
  2. role of Notch signaling in the angiogenesis process.
  3. Why targeting D114 could be advantageous for slowing tumor development.

However, there are a number of substantial issues that need to be resolved, listed below.

  1. Since this overview is not confined to the development of endometrial cancer, the title can be generalized to include a discussion of cancer in total.
  2. If that is the case, I would suggest the authors to include a couple of more pieces of research that the authors may have overlooked in their review.

Wang, H., Huang, X., Zhang, J., Shao, N., ong Chen, L., Ma, D., & Ji, C. (2014). The expression of VEGF and Dll4/Notch pathway molecules in ovarian cancer. Clinica chimica acta436, 243-248.

Liu, Z., Fan, F., Wang, A., Zheng, S., & Lu, Y. (2014). Dll4-Notch signaling in regulation of tumor angiogenesis. Journal of cancer research and clinical oncology140(4), 525-536.

Yen, W. C., Fischer, M. M., Hynes, M., Wu, J., Kim, E., Beviglia, L., ... & Hoey, T. (2012). Anti-DLL4 has broad spectrum activity in pancreatic cancer dependent on targeting DLL4-Notch signaling in both tumor and vasculature cells. Clinical Cancer Research18(19), 5374-5386.

Zohny, S. F., Zamzami, M. A., Al-Malki, A. L., & Trabulsi, N. H. (2020). Highly expressed DLL4 and JAG1: their role in incidence of breast cancer metastasis. Archives of medical research51(2), 145-152.

Etc.

  1. Since the authors are not specifically covering endometrial cancer in this review article, they should include current research that has primarily focused on D114 inhibition and has shown the effect on different cancer stages. Such as,

Jenkins, D. W., Ross, S., Veldman-Jones, M., Foltz, I. N., Clavette, B. C., Manchulenko, K., ... & Barry, S. T. (2012). MEDI0639: a novel therapeutic antibody targeting Dll4 modulates endothelial cell function and angiogenesis in vivo. Molecular cancer therapeutics11(8), 1650-1660.

Yeom, D. H., Lee, Y. S., Ryu, I., Lee, S., Sung, B., Lee, H. B., ... & You, W. K. (2021). ABL001, a Bispecific antibody targeting VEGF and DLL4, with chemotherapy, synergistically inhibits tumor progression in xenograft models. International Journal of Molecular Sciences22(1), 241.

Chiorean, E. G., LoRusso, P., Strother, R. M., Diamond, J. R., Younger, A., Messersmith, W. A., ... & Jimeno, A. (2015). A phase I first-in-human study of enoticumab (REGN421), a fully human delta-like ligand 4 (Dll4) monoclonal antibody in patients with advanced solid tumors. Clinical Cancer Research21(12), 2695-2703.